# Downregulation of hepatic lncRNA Gm19619 improves gluconeogenesis and lipogenesis following vertical sleeve gastrectomy in mice

Zhipeng Fang [1,5], Mingjie Fan[1,2,5], Dongqiang Yuan[1], Lihua Jin[1], Yangmeng Wang[1], Lili Ding[1], Senlin Xu[1,3], Jui Tu[1,3], Eryun Zhang[1], Xiwei Wu[4], Zhen Bouman Chen [1,3] & Wendong Huang [1,3✉]

Long non-coding RNAs (lncRNAs) are emerging important epigenetic regulators in metabolic processes. Whether they contribute to the metabolic effects of vertical sleeve gastrectomy (VSG), one of the most effective treatments for sustainable weight loss and metabolic improvement, is unknown. Herein, we identify a hepatic lncRNA *Gm19619*, which is strongly repressed by VSG but highly up-regulated by diet-induced obesity and overnight-fasting in mice. Forced transcription of *Gm19619* in the mouse liver significantly promotes hepatic gluconeogenesis with the elevated expression of G6pc and Pck1. In contrast, AAV-CasRx mediated knockdown of *Gm19619* in high-fat diet-fed mice significantly improves hepatic glucose and lipid metabolism. Mechanistically, *Gm19619* is enriched along genomic regions encoding leptin receptor (Lepr) and transcription factor Foxo1, as revealed in chromatin isolation by RNA purification (ChIRP) assay and is confirmed to modulate their transcription in the mouse liver. In conclusion, Gm19619 may enhance gluconeogenesis and lipid accumulation in the liver.

[1] Department of Diabetes Complications and Metabolism, Arthur Riggs-Diabetes and Metabolism Research Institute, Beckman Research Institute, City of Hope National Medical Center, Duarte, CA 91010, USA. [2] Department of Pediatric, The First Affiliated Hospital, School of Medicine, Zhejiang University, Hangzhou, Zhejiang 310003, China. [3] Irell & Manella Graduate School of Biological Science, Beckman Research Institute, City of Hope National Medical Center, Duarte, CA 91010, USA. [4] Integrated Genomic Core, City of Hope National Medical Center, Duarte, CA 91010, USA. [5] These authors contributed equally: Zhipeng Fang, Mingjie Fan. ✉email: WHuang@coh.org

Vertical sleeve gastrectomy (VSG) is currently one of the most popular and efficacious bariatric surgeries for sustainable weight loss and the subsequent improvement of diabetes and other obesity-related complications[1,2]. Previously, we and other groups have reported several molecular mechanisms that contribute to the effects of VSG[3–7]. Briefly, we reported that the membrane-bound G protein-coupled bile acid receptor TGR5 was required to maintain the metabolic effects of VSG in mice[5,8]. The expression of TGR5 was increased significantly by VSG to improve glucose control and increase energy expenditure. Recently, we revealed that the intestinal levels of bile acids were significantly reduced by VSG, thereby restricting fat absorption in the intestine[6,9]. However, the molecular mechanism underlying the metabolic improvement of VSG is still largely unknown. A better understanding of the molecular mechanism of VSG may lead to the development of novel therapeutic treatments for obesity and avoid invasive surgery[10,11].

Besides their protein-coding counterparts, long non-coding RNAs (lncRNAs) have been reported to play important roles in metabolic regulation. For example, activation of the liver X receptors (LXR) can robustly induce the expression of a hepatic lncRNA *LeXis*, which then affects the expression of genes involved in cholesterol biosynthesis and alters the cholesterol levels[12]. Cholesterol is also reported to induce other liver lncRNAs, such as *CHROME*, to promote cholesterol efflux and high-density lipoprotein (HDL) biogenesis by binding to several functional microRNAs[13]. Moreover, the lncRNA *lncOb*, which is found to be important for leptin expression, is repressed in diet-induced obese mice and can lead to a leptin-responsive form of obesity[14]. However, little is known whether lncRNAs mediate the metabolic effects of bariatric surgery.

In this study, we took advantage of our well-established VSG mouse models to screen for candidate lncRNAs whose hepatic expression was responsive to VSG operation in mice. Among them, we identified and characterized the functions of a hepatic nuclear lncRNA *Gm19619*, which is repressed by VSG in mice. Interestingly, *Gm19619* was strongly induced by both diet-induced obesity and overnight-fasting. Forced transcription of *Gm19619* by AAV in the mouse liver significantly enhanced hepatic gluconeogenesis. In contrast, AAV-CasRx mediated knockdown of *Gm19619* in HFD-fed mice profoundly improved both hepatic glucose and lipid metabolism. We further demonstrated that *Gm19619* may promote gluconeogenesis by activating the *Foxo1-G6pc/Pck1* pathway, and facilitate lipid accumulation by inhibiting the leptin receptor (*Lepr*) signaling pathway.

## Results

**VSG strongly suppressed the transcription of *Gm19619*.** In our previous studies, we had succeeded in performing the VSG operation on HFD-induced obese mice[5,6]. The results showed that VSG could induce a significant weight loss in mice when compared with sham-operated mice. Blood glucose regulation was also strikingly improved in VSG-operated mice. The RNAseq result confirmed the repression of several key genes involved in lipogenesis, such as *Fasn*, *Scd1*, *Cd36*, and *Srebf1*, and two gluconeogenesis marker genes *G6pc* and *Pck1* (Fig. 1a). These data suggested that the VSG on mice faithfully recapitulates its metabolic effects in human patients. Due to the emerging important roles of lncRNA in metabolism, we profiled the differentially transcribed liver lncRNAs between VSG- and sham-operated mice as well. The results showed around 1900 lncRNAs were mapped to the mouse genome and 40 lncRNAs were found to be significantly changed (fold change >1.67 and $p < 0.05$) after VSG operation in mice (Fig. 1a). Among them, *Gm19619* was verified by qPCR to be a top candidate that was strongly

downregulated in the livers of VSG-operated mice. Moreover, we further confirmed that the hepatic expression level of *Gm19619* in lean mice was much lower compared to that in diet-induced obese mice (Fig. 1b). Additionally, in the overnight-fasted mice, fasting strongly elevated the transcriptional level of *Gm19619* (Fig. 1c). In mouse immortal hepatocytes AML12 cell line, mimicking the activation of glucagon pathway by forskolin treatment strongly elevated the transcription of *Gm19619* (Fig. 1d). All these results consistently indicate that *Gm19619* is a highly responsive lncRNA to VSG and metabolic conditions, which highlights its potential function in regulating hepatic metabolism.

For further study, the comparative genomics method PhyloCSF was first applied to check that *Gm19619* only possessed a barely translational potential[15]. The data from the FANTOM5 database indicated that *Gm19619* was specifically transcribed in mouse liver but no other tissues (Supplementary Fig. 1)[16]. Both the 5′ and 3′ ends of the *Gm19619* transcript were determined by the rapid amplification of cDNA ends PCR (RACE-PCR)[17,18]. By using the primers targeting both ends of *Gm19619* full-length RNA and mouse hepatic cDNA as templates for PCR, we confirmed that *Gm19619* has only one isoform around 1600 nt in mouse liver with a 3′ poly(A) tail (Fig. 1e and Table 1). Because the cellular location is highly correlated with the function of lncRNA, the cytosolic and nuclear RNA in mouse primary hepatocytes were fractionated respectively for quantification. Most of the *Gm19619* RNA was confirmed to be in the nucleus, indicating its potential regulatory function in chromatin gene transcription (Fig. 1f).

**Forced expression of *Gm19619* enhanced hepatic gluconeogenesis.** Because fasting significantly upregulated the transcription of *Gm19619*, we asked whether the forced expression of *Gm19619* RNA would affect the hepatic glucose metabolism in chow-fed mice. To this end, the adeno-associated viruses (AAV) for liver-specific TBG promoter-driven *Gm19619* transcription was constructed, purified, and intravenously injected into mice. GFP was used as a negative control (Fig. 2a). As expected, the AAV strongly increased the hepatic *Gm19619* RNA level (Fig. 2b). In addition, fasting can further elevate the RNA level of *Gm19619* when compared with the re-fed group in both AAV-*Gm19619* and AAV-*GFP* mice (Fig. 2b). The mRNA levels of both *G6pc* and *Pck1* were significantly upregulated after overnight fasting. Interestingly, the forced transcription of *Gm19619* could further promote the increase of both *G6pc* and *Pck1* mRNAs (Fig. 2b). The western blot assays indicated that while there was no significant difference between the expression levels of these two genes in fasted and re-fed group mice, *Gm19619* significantly increased the expression levels of both G6PC and PCK1 in both groups of mice (Fig. 2c). Consistent with the expression of these genes, the fasting blood glucose concentration was significantly upregulated after *Gm19619* forced transcription (Fig. 2d). Glucose tolerance test (GTT) and insulin tolerance test (ITT) results further confirmed that elevation of *Gm19619* transcription impaired the glucose metabolism and insulin sensitivity in mice (Fig. 2e, f), while the pyruvate tolerance test (PTT) result verified that *Gm19619* could remarkably enhance gluconeogesis in the liver (Fig. 2g), although both the body weight and liver/body weight ratio were comparable between these mice (Supplementary Fig. 2a, b). H&E staining showed no detectable morphology changes in either the liver, inguinal white adipose tissue (iWAT), or epididymal white adipose tissue (eWAT) after *Gm19619* forced transcription (Supplementary Fig. 2c). The mRNA level of lipogenic genes, including *Fasn* and *Scd1*, remained unchanged (Supplementary Fig. 2d). These results suggest that *Gm19619* may

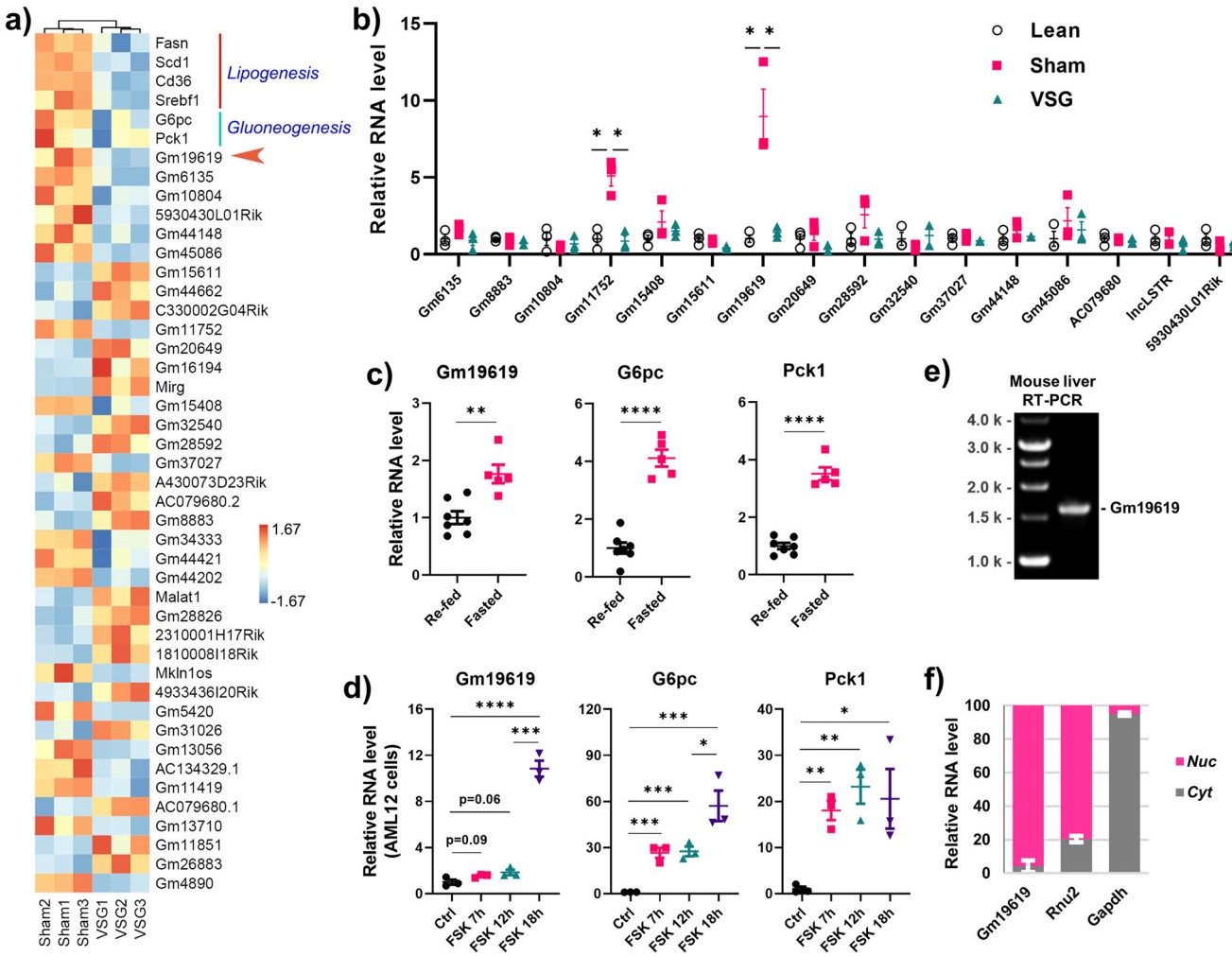

**Fig. 1 *Gm19619* is a hepatic nuclear lncRNA which can be upregulated in both obese-sham-operated and overnight-fasted mice livers. a** Heatmap showing the several protein-coding metabolic genes involved in lipogenesis (*Fasn*, *Scd1*, *Cd36*, and *Srebf1*) and gluconeogenesis (*G6pc* and *Pck1*), and the top 40 differentially transcribed lncRNAs with a fold change over 1.67 ($p < 0.05$) of the counts in FPKM in the RNAseq data of sham or vertical sleeve gastrectomy (VSG) operated mice livers ($n = 3$). **b** The relative level of different lncRNAs in lean, sham, or VSG-operated mice liver by real-time PCR ($n = 3$). **c, d** The relative RNA level of *Gm19619*, *G6pc*, and *Pck1* in overnight-fasted (16 h) ($n = 5$) and re-fed (4 h) ($n = 7$) mice liver (**c**), and in murine immortal hepatocytes AML12 cells after treatment of 10 μM forskolin for 7, 12, and 18 h ($n = 3$) (**d**). **e** The full length of *Gm19619* RNA was determined by RT-PCR using mouse liver cDNA after its 5′ and 3′ sequence was confirmed by RACE-PCR. **f** Real-time PCR determined the distribution of *Gm19619* RNA in nucleus and cytosol ($n = 3$). *Rnu2* and *Gapdh* RNA served as the marker of the nucleus and cytosol, respectively. Two-tailed unpaired *t*-test. Error bars represent the SEM. *$P < 0.05$, **$P < 0.01$, ***$P < 0.001$, and ****$P < 0.0001$.

promote hepatic gluconeogenesis but not affect lipid metabolism under normal conditions.

**Knockdown of *Gm19619* in the HFD-fed mice inhibited the gluconeogenesis.** The highly elevated levels of *Gm19619* in both obese mice and overnight-fasted mice and its potential role in promoting hepatic gluconeogenesis suggest that it may also be involved in the pathogenesis of diet-induced obesity and diabetes. To determine its potential function in obesity, we constructed the AAV-based vector containing the CasRx gene with concatenated nuclear localization signal peptides to specifically knockdown *Gm19619* lncRNA in the nucleus. Three concatenated sgRNAs targeting *Gm19619* spaced by DR36 sequence were inserted and transcribed under the human U6 promoter (Fig. 3a). The quantitative PCR results indicated the intravenously injected AAV efficiently silenced the hepatic *Gm19619* transcription in the HFD-fed mice (Fig. 3b). As expected, knockdown of *Gm19619* RNA significantly lowered the mouse fasting blood glucose level (Fig. 3c), improved the glucose tolerance (GTT assay) and the

insulin resistance (ITT assay) of HFD-fed mice (Fig. 3d, e). Moreover, the PTT assay further confirmed that *Gm19619* knockdown repressed gluconeogenesis in mice (Fig. 3f). Accordingly, the mRNA level of the *Pck1* gene was significantly decreased, although *G6pc* remained unchanged (Fig. 3b). The protein levels of G6PC and PCK1 were found to be consistent with their transcriptional changes (Fig. 3g). These results indicated that knockdown of *Gm19619* could significantly ameliorate glucose intolerance and insulin resistance in HFD-fed mice.

**Knockdown of *Gm19619* in the HFD-induced obese mice improved lipid metabolism.** Although knockdown of *Gm19619* showed no significant effect on the body weight when compared with the control group (Fig. 4a), H&E staining and oil red o staining showed smaller droplets and less lipid accumulation in *Gm19619* knockdown group (Fig. 4b), suggesting its role in hepatic lipid metabolism under obesity condition. The hepatic triglycerides (TG) levels were confirmed to be strongly decreased after *Gm19619* knockdown (Fig. 4c), while the amount of serum

**Table 1 RNA sequence of *Gm19619* without a polyA tail.**

>*Gm19619*_RNA_1593nt
CCCACCCAGGAACAGAGACUUUAGGGUGCUUCAAGCAAGAGGGGAGUGUAACCAAGGACUGCCACUUCACUCUCCUUCAUAUGCUUAGCCGAAC
UUAGGAAAGGAACUGCAGGGAUGGAAAUGAAGAGGAGCCUGAGGAAAUGAAGGUCCAAGGACAGGCCCAAAGUAGGAUCCAGCUCAAGGGGA
GGCCCCAAGGCCUGACACUAUUACUGAGGCUAUGGAGCGUUCACAAAAAGGGACCCUGUCAUGGCCACACUCCAGAAGACCCAACAAGAAGCU
GAAAGAGACUGGGCAUUCUAUUGGGCUCAGUGUCUCCCAUCUUCUGAUGAAACCAGGAUCACAUUAUUCUUAAGAAUAUGAUAGAGGGCAAGCU
UAAGCGGAAUCAGACAAGGUUCUCUGCUUUCACAUGGCACAGCUCCCAGUGUUUGGAAACAGAGCCUUUCCUAUUCCAGGCCAGGAUCUCGUC
UGGCCUGCAGUUAGAGGCAUCAUACUCAGCUCACACCCUCCGGUGUGUCAAUGACCUUGAAAUACAAUGCAGAGAACGUAAGAGAUUCUUCUU
GACAGUGACCAUCCUCGAUCCAGAAAGCAUACUUGGAGAAUACCAAGUCCGGGUCUUUUCAUCAACUCGAGGUCAACUCAAUUUCCAUAUCAUC
UGGUGUGUCUUAGGGGUAUCCAGGACUUGGGUGUUGUGGGACAACUGGGUUCUGAUGAUGCCAAAUAGCCUUGGCUUCUGUUGAUGGUAUUGGAAGU
GGAAAGAAGAACUCAUGUAGCAGUAGUGUCUUGGCUCCCAGGAAAACACAAUGACAGGAAAGACAAAGGCUGUCACUACUACUGUGCUUAUUAC
UCCUACAUCUCACAGCUCAGUUCCCCUGUCCUUUUAACUUCUCGUUUUUUCAUAAAUUGUGACACCUCUGAAGUAAUGGUCAUUUGGAUUGUUUUA
UUUCAUCACUAUCCACAAUCUGUAAUCUACAAAAGUCUUCAUUAUUGACAAGUGACCUUUGUUUAAUCUUUAUGAAUUAAAAAUCUACAAACAAAG
GAGCUGUCCAGAACCCAUAGUUACAUUAGAAAAAAACACACACAAAAGAAUGGAAUUCAACUCAGAGGGAAUGGUGGCUUCACCCAGUCUCUU
UUAAUGAAACUACUCCUUCGGAAAUGUGCUUGCCCUAACACAGCACUUGGCCUGCAAACAGUGGCACUAUUAAUGUAAAUUAUUCCAGUCUUGU
GGUUUGAUAGAAAGAUUAAUGGUCUCGAAGCUCAAAGAUAAAUGCUGUGCUUGUGUAAUGAGGUCAACAAUCACCAAAGGCCACUGGAGGCCUC
UGACAGGGAAAAACCAGUGGUUAAGAUCUGUAUGUUGUUAGCUCAUCUGGGUGAUGUGACCCUGGACUCUGAUGCUUAGAAAGCUAGCACCAG
CUACUGAAUCUGCUAAUUAACUCUAGCAGCUUCCAGGUUCUCUUGGGACAUCGUAUCAUCUGUCCCUCUUAAAAAAAUUAAAAACUGAACCACAU
CAGUAGAACUCAAAAGAGAUAAGAUCUACCUUUAAUAAUACUCCAUUAAACUAUAUACUCCUCCUGCUCUUUGUUCUAAUAUAUAUGACAGUA
UGCCACU

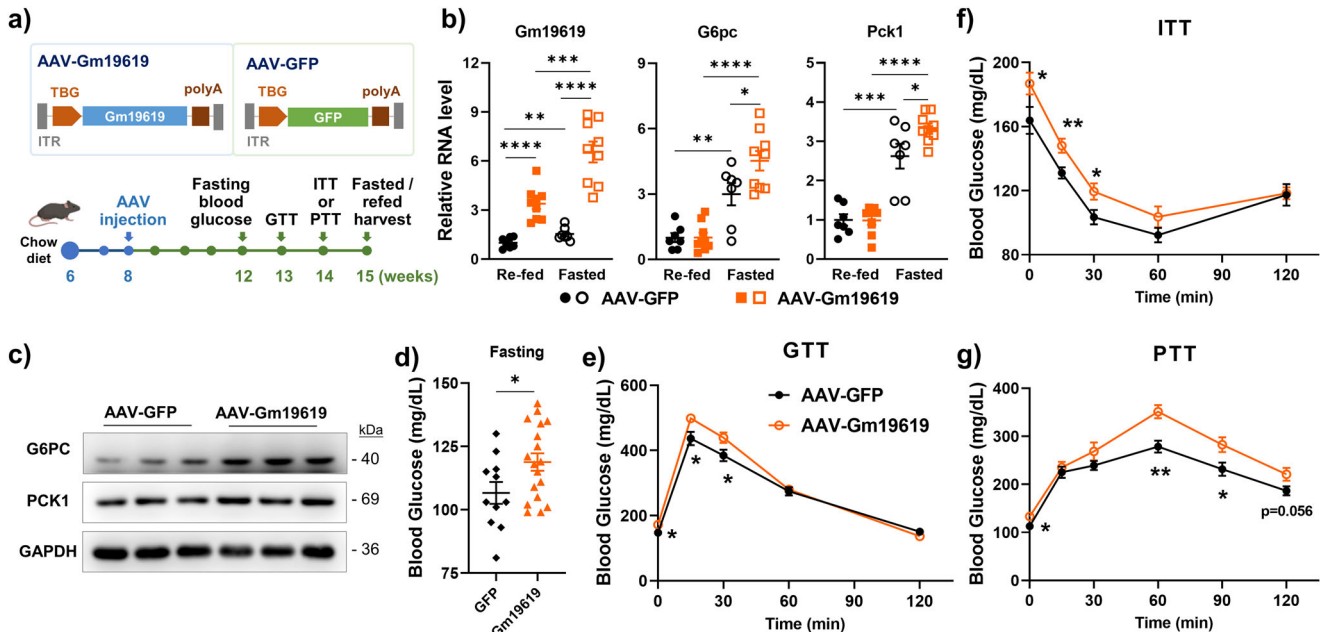

**Fig. 2 Adenovirus-associated virus (AAV) mediated hepatic *Gm19619* forced transcription promoted hepatic gluconeogenesis in chow-fed mice.**
**a** Schema of the AAV vector for hepatic *Gm19619* transcription under the liver-specific TBG promoter and the experimental timeline. *GFP* was used as the control. **b** The relative RNA level of *Gm19619*, *G6pc*, and *Pck1* in AAV-*Gm19619* (*n* = 9) or AAV-*GFP* (*n* = 7) infected mice liver after 16 h fasting and 4 h refeeding by real-time PCR. **c** Western blot analysis of protein levels of G6PC and PCK1 in AAV-*Gm19619* and AAV-*GFP* mice liver (*n* = 3). **d** The blood glucose level of AAV-*Gm19619* (*n* = 18) and AAV-*GFP* (*n* = 11) mice after overnight fasting. **e-g** GTT (**e**) (*n* = 11), ITT (**f**) (*n* = 11), and PTT (**g**) (*n* = 8) assay of the AAV-*Gm19619* and AAV-*GFP* mice. Two-tailed unpaired *t*-test. Error bars represent the SEM. *$P < 0.05$, **$P < 0.01$, ***$P < 0.001$, and ****$P < 0.0001$.

non-esterified fatty acid (NEFA) was significantly upregulated (Fig. 4d). These changes were accompanied by the decreased mRNA levels of several lipogenesis genes such as *Srebf1*, *Fasn*, *Scd1*, and the fatty acid absorption receptor *Cd36* (Fig. 4e). Moreover, the transcription of several genes mediating lipolysis and fatty acid beta-oxidation, including *Acsl1*, *Pnpla2*, *Lipe*, *Cpt1a*, and *Cpt1b*, were elevated by *Gm19619* knockdown (Fig. 4f), indicating *Gm19619* can promote lipogenesis and inhibit lipolysis pathways in the HFD-fed mice liver.

**Gm19619 downregulated hepatic metabolic signaling pathways.** To investigate the potential mechanism by which *Gm19619*

regulates hepatic metabolism, we first tested if the transcriptional level of its two nearby genes *Areg* (Amphiregulin) and *Btc* (Betacellulin), was correlated with *Gm19619* RNA, as many lncRNAs function by modulating their nearby gene level[19]. The real-time PCR data indicated that *Areg* was barely transcribed in the liver and not correlated with Gm19619 (Supplementary Fig 3a), while the *Btc* RNA level was much higher in sham mice liver than VSG mice liver, which was like *Gm19619*. However, the forced transcription of *Gm19619* did not affect the transcription of *Btc* in the liver (Supplementary Fig. 3b), indicating that *Gm19619* may not function by regulating its nearby gene expression.

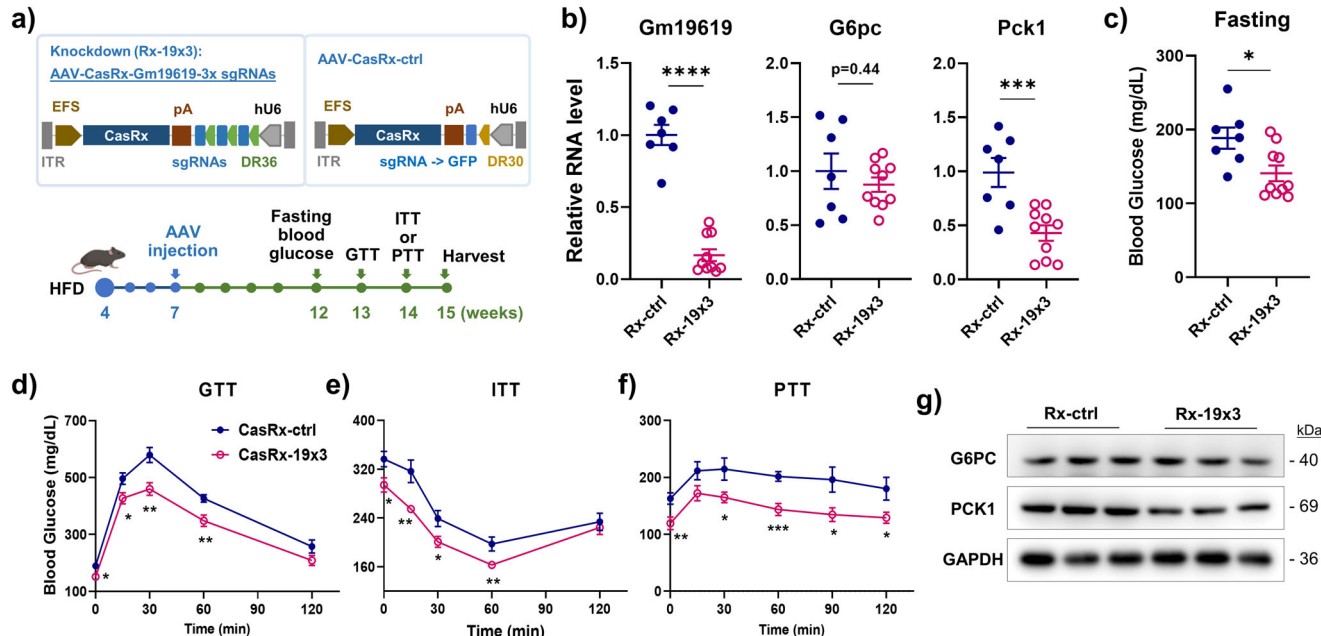

**Fig. 3 AAV-CasRx mediated *Gm19619* knockdown in high-fat diet (HFD) fed mice significantly improved the mice glucose metabolism. a** Schema of the AAV-EFS-CasRx vector for *Gm19619* knockdown and the experimental timeline. **b** The relative RNA levels of hepatic *Gm19619*, *G6pc*, and *Pck1* in *Gm19619* knockdown (CasRx-19×3) ($n = 10$) and control (CasRx-ctrl) ($n = 7$) mice. **c** The fasting blood glucose levels in CasRx-19×3 mice ($n = 10$) and CasRx-ctrl ($n = 7$) mice. **d-f** The GTT (**d**), ITT (**e**), and PTT (**f**) assay of CasRx-19×3 ($n = 10$, PTT $n = 8$) and CasRx-ctrl ($n = 7$) mice. **g** The western blot results of G6PC and PCK1 in CasRx-19×3 and control group mice livers ($n = 3$). Two-tailed unpaired *t*-test. Error bars represent the SEM. *$P < 0.05$, **$P < 0.01$, and ***$P < 0.001$.

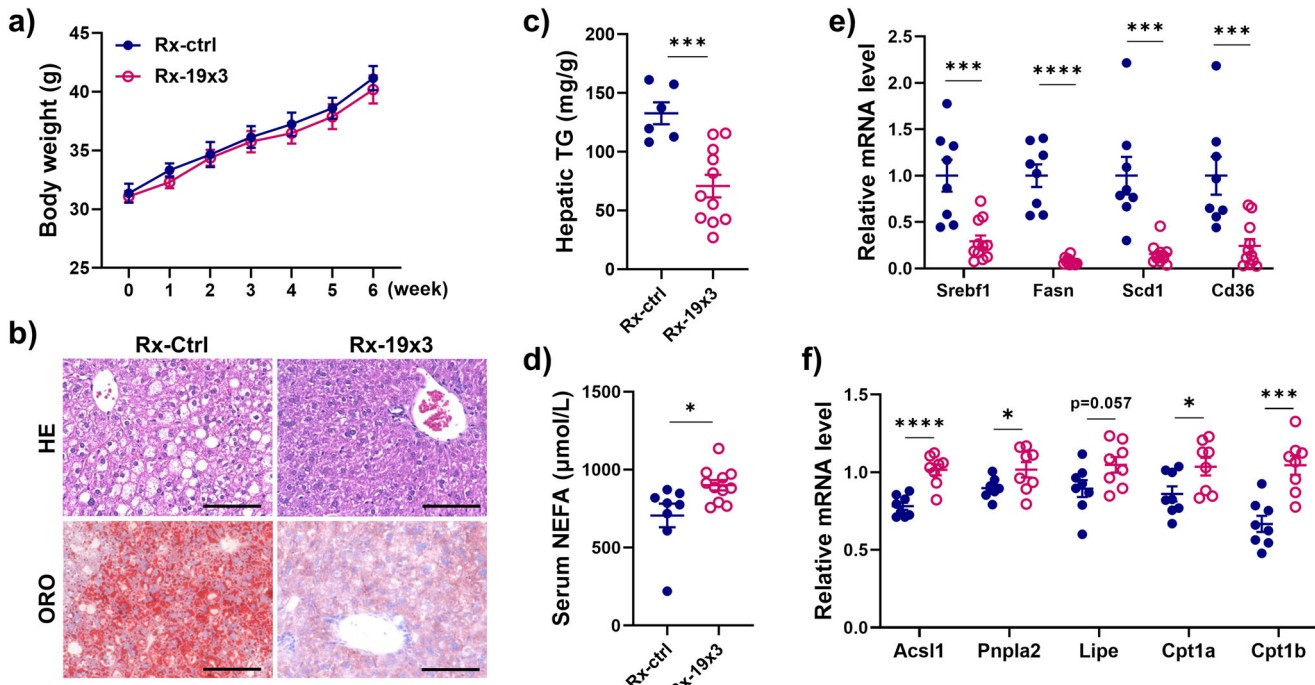

**Fig. 4 AAV-CasRx mediated *Gm19619* knockdown in HFD-fed mice significantly improved the mice lipid metabolism. a** The body weight of *Gm19619* knockdown (CasRx-19×3) ($n = 11$) and control ($n = 8$) mice. **b** Representative images of liver sections by H&E and oil red o staining, respectively. Scale bar, 100 μm. **c** The serum level of triacylglycerol (TG) in CasRx-19×3 ($n = 11$) and CasRx-ctrl ($n = 6$) mice. **d** The serum level of NEFA in CasRx-19×3 ($n = 11$) and CasRx-ctrl ($n = 8$) mice. **e** The relative mRNA levels of some lipogenesis genes and *Cd36* in CasRx-19×3 ($n = 11$) and CasRx-ctrl ($n = 8$) mice. **f** The relative mRNA levels of several lipolysis and beta-oxidation genes in CasRx-19×3 and CasRx-ctrl mice ($n = 8$). Two-tailed unpaired *t*-test. Error bars represent the SEM. *$P < 0.05$, ***$P < 0.001$, and ****$P < 0.0001$.

For the further mechanistic study, we performed chromatin isolation by RNA purification (ChIRP) assay by using freshly fixed primary murine hepatocytes from wild-type mice. Fourteen probes with antisense to different regions of *Gm19619* RNA were divided into two sets (odd- and even-numbered) for the assay (Fig. 5a and Supplementary Table 1). The pull downed chromatin DNA was sequenced and mapped to the mouse genome[20]. The analysis of the mapped DNA sequences indicates that most of *Gm19619* RNA bind to the distal intergenic regions and introns, while 9% of the binding region was within the promoter region (Fig. 5b). The pathway enrichment analysis of genes showing interactions with *Gm19619* revealed that leptin receptor signaling was among the top potential pathways affected by *Gm19619* (Fig. 5c, d). While *Gm19619* showed binding to its own genomic locus (Fig. 5e), ChIRP-seq also revealed the *Gm19619* binding to the upstream region and genomic locus of *Lepr* (Fig. 5f), as well as *Plcg1*, *Adcy8*, *Prkar2a*, and *Npy* (Supplementary Fig. 4). The function of leptin receptor signaling in obesity pathogenesis has been well reported, in part supported by the striking metabolic phenotype exhibited by the db/db mouse which carries *Lepr* mutation[21].

QPCR data showed the transcriptional level of *Lepr* was strikingly induced in overnight-fasted mice when compared with the re-fed mice (Fig. 5g), and AAV-Gm19619 significantly reduced the fasting-induced *Lepr* mRNA level. However, western blotting showed that fasting suppressed the long active isoform of LEPR (~125 kDa) expression, and AAV-*Gm19619* did not affect the LEPR expression in both fasting and feeding status (Fig. 5h), consistent with the negligible effect of *Gm19619* on hepatic lipid metabolism in chow-fed mice (Supplementary Fig. 2). In contrast, the *Gm19619* knockdown highly upregulated both the mRNA and protein levels of LEPR in HFD-fed mice (Fig. 5i, j). Similarly, the transcriptional repression of *Plcg1* by fasting was interrupted in AAV-Gm19619 mice (Fig. 5g); although the knockdown of *Gm19619* did not affect the mRNA level of *Plcg1* (Fig. 5i). These results suggest that *Gm19619* may repress the leptin receptor pathway in the liver, which contributed to its regulation of hepatic lipid metabolism mainly under the obese condition.

ChIRP data also identified that *Gm19619* could bind to the promoter region and genomic locus of *Foxo1*, a key transcription factor (TF) in gluconeogenesis (Fig. 5k), suggesting that *Foxo1* may also be a potential downstream effector of *Gm19619* to promote the gluconeogenesis in the liver. Consistent with a recent report[22], the protein expression of FOXO1 was found to be repressed in fasted mouse liver when compared with that in the re-fed mouse liver (Fig. 5h). In the fasted mice, AAV-*Gm19619* significantly increased both the mRNA and protein levels of hepatic Foxo1, but not Creb1 (Fig. 5g, h). On the other hand, *Gm19619* knockdown in HFD-fed mice repressed the expression of FOXO1 (Fig. 5i, j), which might contribute to the improved glucose metabolism in HFD-fed mice.

In summary, our data show that VSG in mice significantly represses the transcription of hepatic *Gm19619*, while both obesity and fasting induce its upregulation. *Gm19619* lncRNA may promote gluconeogenesis by activating the *Foxo1* pathway, and facilitate lipogenesis and repress lipolysis by inhibiting the hepatic leptin receptor signaling (Fig. 6).

## Discussion

VSG is currently one of the most common and effective treatments for the sustained weight loss and remission of diabetes, as well as other comorbidities such as nonalcoholic fatty liver disease (NAFLD). However, it is still an invasive surgery with low mortality probability and may not be suitable for some specific groups of people. A better understanding of the mechanism

underneath the beneficial effect of VSG or other bariatric surgeries may lead to the development of less invasive and cost-effective therapeutic approaches. In this study, we used VSG-operated mice and fasting/refeeding mice as models and found that the obesity and fasting-induced hepatic lncRNA *Gm19619* was highly suppressed after VSG operation. Forced transcription of *Gm19619* activated hepatic gluconeogenesis, while AAV-CasRx mediated in vivo hepatic knockdown significantly improved the glucose and lipid metabolism in HFD-fed mice.

Mechanically, *Gm19619* may affect several metabolic signaling pathways that regulate hepatic glucose and lipid metabolism. Among them, the hepatic leptin receptor signaling pathway is a major downstream mediator of *Gm19619*. Leptin receptor signaling is particularly important in obesity pathogenesis. Some recent studies showed that mere ablation of hepatic leptin receptors could promote lipid accumulation in the liver and produce more enlarged triglyceride-rich VLDL particles[23,24]. Our results confirmed that the leptin receptor pathway was regulated by *Gm19619*, particularly in the HFD-fed mice, indicating that *Gm19619* might promote lipid accumulation by repressing the leptin receptor pathway in the liver under the obese condition. Surprisingly, the circulating leptin level was found to be reduced after VSG[25], and yet VSG can still reduce fat mass and improve glucose tolerance in *ob/ob* mice in the absence of the leptin signaling[26]. In this regard, in addition to the repression of *Gm19619* in the liver, other signaling pathways may also have a significant impact on lipid metabolism after the VSG surgery.

Moreover, a recent study reported that activation of hepatic leptin signaling improved hyperglycemia by degrading MARK phosphatase-3 (also known as dual specificity phosphatase 6, DUSP6) and subsequent repressing *Foxo1*-mediated gluconeogenesis pathway[27], indicating *Gm19619* may promote the hepatic gluconeogenesis by repressing the hepatic leptin signaling. Indeed, we found that *Gm19619* also bound to the promoter of the *Foxo1* and activated its expression to modulate the hepatic glucose metabolism. Thus, the downregulation of *Gm19619* by VSG may contribute to the improved hepatic glucose metabolism in part through the *Foxo1* pathway. The distinct consequence of Gm19619 binding in different genes (e.g., *Lepr* vs *Foxo1*) will warrant future investigation.

Besides *Gm19619*, we identified other hepatic lncRNAs differentially expressed after VSG surgery. For example, a lncRNA *Gm10680*, which was also significantly repressed by VSG, has been reported to be involved in hepatic lipid metabolism. Knockdown of this lncRNA reduced the nearby *APOA4* gene expression and led to reduced levels of plasma triglyceride and total cholesterol in *ob/ob* mice[28]. It is likely that *Gm10680* and other lncRNAs involved in the various metabolic pathways can collectively lead to metabolic improvement after VSG surgery. In addition to the liver lncRNAs, lncRNAs such as *GAS5* in adipose tissues, small intestine, and other tissues may also contribute to the overall metabolic improvement by VSG[29,30].

LncRNAs can recruit RNA-binding proteins and transcription factors to regulate downstream gene function. To identify potential binding proteins of *Gm19619*, we analyzed the binding motifs and TF binding sites enriched in the binding sequences of *Gm19619*. This analysis revealed that *Gm19619* could likely interplay with several TFs, including SREBF1, SREBF2, FXR, LXR, and HNF4α, to regulate hepatic glucose and lipid metabolism (Supplementary Fig. 5). Moreover, the prediction of potential TFs that may bind to and regulate *Gm19619* promoter activity (by the JASPAR[31]) showed that *Gm19619* may be regulated by CREB1, PPARα, FXR, and additional TFs known to regulate glucose metabolism in fasted and fed states (Supplementary Fig. 6)[32]. A more comprehensive investigation of the metabolic modulation of *Gm19619* will be our future endeavor.

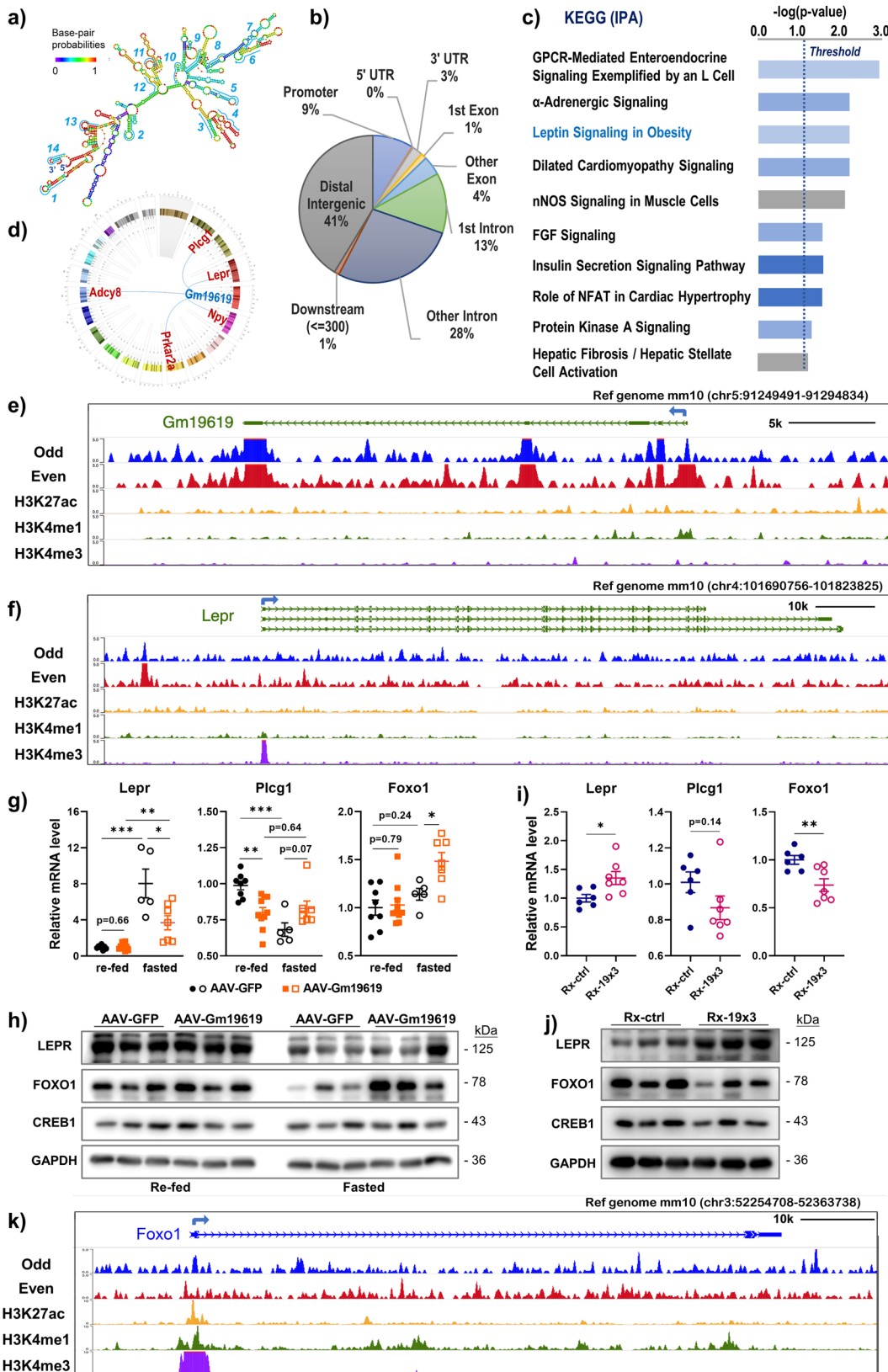

We used AAV-CasRx to knock down the hepatic RNA of *Gm19619*. Compared with CRISPR-Cas9 mediated chromatin DNA disruption or CRISPRi mediated transcriptional repression[33], CasRx will only bind and cut RNA[34], without any interaction with chromatin DNA. CRISPR-Cas9 with a single guide RNA for gene knockout does not work for lncRNA, as a small insertion or deletion may not affect the function of most lncRNAs. Therefore, pair of sgRNAs is always applied to knock out a large DNA fragment in the lncRNA locus to totally abolish its transcription[35]. However, this technique may fail to dissect if the disrupted function of lncRNA is due to the transcript itself or the DNA elements, such as enhancers on the chromosome.

**Fig. 5 ChIRP data analysis indicated Gm19619 binds to the chromatic DNA of leptin receptor and Foxo1 and modulates their expression. a** The second structure of *Gm19619* RNA with minimal free energy as predicted on the RNAfold website. The locations of the 14 ChIRP probes were shown. **b** The genomic distribution of ChIRP pulldown DNA fragments with a pileup over 5 and longer than 200 bp. **c** Pathway enrichment analysis of genes with surrounding regions bound by *Gm19619* identified from ChIRP-seq. The gene list from overlapping peaks was used to run KEGG pathway analysis with IPA (Ingenuity Pathway Analysis). **d** The Circos plot showing the ChIRP enriched genes within the leptin receptor pathway. **e, f, k** ChIRP-seq signals indicate *Gm19619* binding along its own exon locus (**e**), the leptin receptor *Lepr* (**f**), and the *Foxo1* gene locus (**k**). H3K27ac, H3K4me1, and H3K4me3 signals from ENCODE mouse liver ChIP-seq data were shown as control, mm10 as the reference genome. **g** The relative mRNA levels of *Lepr, Plcg1*, and *Foxo1* in *Gm19619* forced expressed mice ($n = 5$–10). **h** Western blot analysis of protein levels of LEPR, FOXO1, and CREB1 in AAV-*Gm19619* and AAV-*GFP* mice liver ($n = 3$). **i** The relative mRNA levels of *Lepr, Plcg1,* and *Foxo1* in *Gm19619* knockdown mice ($n = 6$–7). **j** The western blotting of LEPR, FOXO1, and CREB1 in the livers of CasRx-19×3 and control mice ($n = 3$). Two-tailed unpaired *t*-test. Error bars represent the SEM. *$P < 0.05$, **$P < 0.01$, and ***$P < 0.001$.

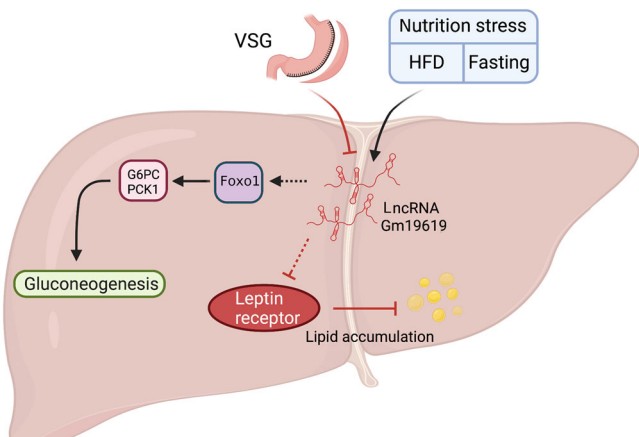

**Fig. 6 Schema showing the potential function of *Gm19619* in the liver.** While VSG represses the transcription of hepatic *Gm19619*, both HFD and fasting induce *Gm19619* RNA. *Gm19619* may promote gluconeogenesis by activating the *Foxo1-G6pc/Pck1* pathway and facilitate lipid accumulation by inhibiting the leptin receptor signaling pathway.

Currently, there are no CasRx transgenic mice available. It can be anticipated that the development of this kind of mice will facilitate the in vivo studies of lncRNAs in the future.

The discovery of lncRNA expands our knowledge of the complexity of vertebrate genomes, but the low sequence conservation among varied species may be a barrier to the understanding of relevant human physiology and disease research. One approach to address this problem may be the use of humanized mice. For example, Ruan et al. used a humanized TK-NOG mouse model where the liver was repopulated with human hepatocytes to investigate the function of non-conserved human hepatic lncRNAs associated with NAFLD and metabolic homeostasis. They managed to identify *LINC01018* as an obesity-associated lncRNA, and *hLMR1* promotes the transcription of cholesterol biosynthetic genes[36,37]. Further effort should be made to test if such humanized mouse models can be used for bariatric surgery studies, which may better recapitulate the situation in human patients. Another alternative approach will be using the human lncRNA knock-in mouse models, although many challenges remain to be solved[38].

## Methods

**Animals**. Wild-type C57BL/6 male mice (4–8 weeks old) were purchased from the Jackson Laboratory (Bar Harbor, ME). All animal procedures were approved by the City of Hope Institutional Animal Care and Use Committee (IACUC protocol: #14031) and conducted in accordance with the National Institutes of Health Guidelines for the Care and Use of Laboratory Animals. Mice were housed in a 12-h light/dark cycle in a temperature-controlled room (25 °C) with a relative humidity of 50%. Animals were given ad libitum access to water and a regular chow diet in a temperature-controlled environment.

**Primary hepatocytes isolation**. Primary hepatocytes were isolated by two-step methods. The anesthetic mouse was perfused with 50 mL 1× DPBS without $Ca^{2+}$ and with 5 mM EDTA through a portal vein at a speed of 5 mL/min, then 25 mg of collagenase IV (Worthington) dissolved in 50 mL of HBSS buffer was perfused. The liver was harvested and rid of the gallbladder. The hepatocytes were shaken off by tweezers into DMEM with 10% FBS and antibiotics and collected by centrifuge at $50 \times g$ for 3 min. The cells were then washed twice with DMEM with 10% FBS. The purified primary hepatocytes were immediately seeded into a collagen-pretreated six-well plate or used for other experiments.

**Primary hepatocytes cytosolic and nuclear RNA isolation**. The nucleus and cytosol of the hepatocytes were isolated by NE-PER Nuclear and Cytoplasmic Extraction Reagents (Thermo Fisher) according to the vendor's instruction. SUPERase•In RNase inhibitor was added into the separation buffers to prevent RNA degradation. The RNA was directly isolated by the Direct-zol RNA kit (Zymo Research) and eluted using the same volume of RNase-free water. About 1 ug cytoplasmic RNA and the same volume of nuclear RNA was reversely transcribed respectively by using the high-capacity cDNA reverse transcription kit from Thermo Fisher.

**RACE-PCR**. The nucleus RNA isolated from murine hepatocytes was used for the 5′ and 3′ ends of *Gm19619* determination by the RACE-PCR according to the published protocols[17,18]. Briefly, for the 5′ RACE, the primer RA5-1 (5′-CTCAGCCAATGATCTCCAAC-3′) was used for reverse transcription, then the RNA template was degraded by RNase H and poly(A) tail was added to first-strand cDNA products by the terminal deoxynucleotidyl transferase. Three primers RA5-2 (5′-GTGACATAGGTCCTTTGGC-3′), Qdt (5′-CCAGTGAGCAGAGTGACGAGGACTCGAGCTCAAGCTTTTTTTTTTTTTTTTTTT-3′), and Qouter (5′-CCAGTGAGCAGAGTGACG-3′) was used for the first round of the PCR. The primers RA 5-3 (5′-GCTTGAAGACTCCTTGCTG-3′) and Qinner (5′-GAGGACTCGAGCTCAAGC-3′) were used for the second round of PCR.

For the 3′ RACE, the primer Qdt was used for the RNA reverse transcription, then primer *Gm19619_QF* (AAGCATCAGAGTCCAGGGTC) and Qouter were used for the first round of PCR. The primer 3RA22-F(5′-ACAGCACTTGGCCTGCAAAC-3′) and Qinner were used for the second round of PCR. The amplicons of these PCR were gel purified and sent for sanger sequencing.

Based on the sequencing results, the primers *Gm19619_nFo* (5′-CCCACCCAGGAACAGAGACTTTA-3′) and *Gm19619_nRo* (5′-AGTGGCATACTGTCATATATATTAGAACAAA-3′) was used for the full-length PCR amplification.

**AAV-TBG-*Gm19619* for lncRNA *Gm19619* transcription in chow-fed mice**. The sequence of *Gm19619* was amplified by the primers *Gm19619_NotI_F* (5′-aGCGGCCGCccacccaggaacagagact-3′) and *Gm19619_Esp3I_R* (5′-tCGTCTCa-GATCagtggcatactgtcatatatattag-3′), digested and replaced the eGFP sequence in pAAV-TBG-eGFP (Addgene #105535). The plasmid was sequenced and then used for AAV packing in 293 T cells according to the AAV protocols at the Addgene website and our previous report[39]. Eight-week-old C57BL/6 mice were received once intravenous injections of $5 \times 10^{11}$ genome copies of AAV-TBG-*Gm19619*, or AAV-TBG-eGFP for control. Animals were fed with the standard chow ad libitum. The mice were euthanized after being fasted for 16 h or a 16 h fasting followed by 4 h refeeding.

**AAV-CasRx mediated knockdown of lncRNA *Gm19619* in HFD mice**. The plasmid of pAAV-EFS-CasRx was constructed by cloning the EFS promoter and *RfxCas13d* from the pLentiRNACRISPR_005 plasmid (Addgene #138147) and replaced the TBG promoter and *SaCas9* gene in px602 plasmid (Addgene #61593). The DR30 sgRNA cassette for CasRx sgRNA cloning was used to further replace the Sa gRNA scaffold. Five individual sgRNA candidates targeting *Gm19619* was inserted after the DR30 respectively and verified their knockdown efficacy in 293 T cells against the co-transfected pCMV-*Gm19619* plasmid by QPCR. Then three sgRNAs were chosen to be concatenated and spaced by DR36 sequences to construct the pAAV-EFS-CasRx-19×3 plasmid for AAV packaging. Four-week-old C57BL/6 mice were fed with HFD for 3 weeks, then received once intravenous

injections of $5 \times 10^{11}$ genome copies of AAV-CasRx-19×3, as well as AAV-CasRx-GFP for control and continued on HFD until euthanization.

**Glucose tolerance test (GTT)/pyruvate tolerance test (PTT)**. Mice were fasted for 14 h prior to GTT or PTT. An AlphaTRAK 2 Blood Glucose Monitoring System was used to measure blood glucose from blood collected by tail nick. After the initial blood glucose reading, mice were intraperitoneally injected with glucose solution at 2 g/kg of body weight, or pyruvate solution at 2.5 g/kg of body weight[40]. Then, blood glucose was measured at 15, 30, 60, 90, and 120 min after glucose or pyruvate injection.

**Insulin tolerance test (ITT)**. Mice were fasted for 6 h prior to ITT. Blood glucose level was measured from blood collected by tail prick with an AlphaTRAK 2 Blood Glucose Monitoring System. After the initial blood glucose reading (Time 0 min), mice were injected with insulin intraperitoneally at 0.75 U/kg of body weight. Blood glucose was measured at 15, 30, 60, and 120 min after insulin injection.

**Serum analysis**. Serum ALT (alanine transaminase) and AST (aspartate transaminase) levels were measured using the EnzyChrom™ Alanine Transaminase Assay Kit (Bioassay, EALT100, Hayward, CA) and the EnzyChrom™ Aspartate Transaminase Assay Kit (Bioassay, EASTR-100, Hayward, CA). Serum levels of non-esterified FFA (NEFA) were measured using a NEFA Assay kit (Wako, TK036, Neuss, Deutschland). Hepatic triglyceride levels were measured using the L-Type Triglycerides M Kit (Wako, TK006, Neuss, Deutschland).

**Hematoxylin and eosin (H&E) and Oil Red O staining**. After mice were euthanized, their tissues were immediately removed, immersed in cold PBS, and fixed in 10% formalin solution for 24–48 h at 4 °C. For H&E staining, the samples were then incubated in 70% ethanol for 12 h, embedded in paraffin, and sectioned into 5 μm sections. For Oil Red O staining of the liver sections, the tissues were fixed in 4% paraformaldehyde, then incubated in 30% sucrose for 12 h and embedded in Tissue-Tek® O.C.T. Compound (Sakura Finetek, Torrance, CA, USA). Serial sections (10 μM) were made and stained with 0.5% Oil Red O for 10 min and counter-stained with hematoxylin. The red lipid droplets were visualized by microscopy.

**Western blot**. The liver lysate samples of 30 μg per lane of gel were used for the SDS-PAGE and western blot. The used antibodies are listed below: G6PC (1:500), Sigma Aldrich, HPA052324-25UL; PCK1 (1:1000), Santa Cruz, sc-74825; LEPR (1:500), Novus Bio, NB120-5593; FOXO1 (1:1000), Cell Signaling, 2880 S; CREB1 (1:1000), Cell Signaling, 9197 S; GAPDH (1:4000), Cell Signaling, 2118 S.

**QPCR**. Total RNA was extracted from tissues and cells using TRI reagent (MRC, TR118, Cincinnati, OH) and reverse transcribed by using Applied Biosystems™ high-capacity reverse transcription kit (Thermo Fisher). Quantitative real-time PCR was performed with SYBR Green PCR Master Mix (Applied Biosystems) using an ABI Vii7a Real-Time PCR System (Applied Biosystems, Waltham, MA). The sequences of primers used are listed in Supplementary Table 2. mRNA levels were normalized to *Rplp0*.

**RNAseq and data analysis**. The liver total RNA from 3 VSG and 3 sham mice was extracted by the Direct-Zol kit (Zymo Research), respectively, and used for the polyA-stranded-RNAseq library preparation. The sequencing was performed by Illumina Hiseq2500 at the City of Hope Integrative Genomics Core. FastQC (version 0.10.1) was used for quality control and STAR (version 2.6.0.a) was used for the alignment to the mouse reference genome (mm10). The read counts were calculated by featureCounts in the Subread package (2.0.1). R (version 4.2.0) and DESeq2 (version 1.37.0) were used for the differential expression analysis. The counts were presented as FPKM (Fragments Per Kilobase Million). The heatmap was drawn by using pheatmap (version 1.0.12). The RNAseq data can be retrieved from the NCBI GEO database (GSE218931).

**ChIRP and data analysis**. About 250 million primary hepatocytes were isolated from four female 8-week-old WT C57BL/6 mice. These hepatocytes were fixed in 90 mL 3% formaldehyde in 1x DPBS for 10 min. Then 10 mL 1.25 M glycine was added to quench the formaldehyde. The hepatocytes were washed by 1x DPBS, and aliquoted by 60 million cells per tube in ice-cold DPBS and stored at −80 °C. The formaldehyde-fixed hepatocytes were sonicated in 0.5 mL standard ChIRP lysis buffer per tube at the setting of 30-s ON / 30-s OFF for 11 cycles, and then centrifuged at 16,100×*g* for 10 min, 4 °C. The supernatant was combined and aliquoted to 1 mL/tube. Fourteen probes with antisense to different regions of *Gm19619* were divided into two sets (odd- and even-numbered) for the assay, 100 μL Dynabead C1 (Thermo Fisher) per sample was used to pulldown the chromatin fragments according to the JOVE protocol[41] and sent for ChIP-seq.

ChIRP-seq was analyzed following the pipeline described by ref.[42]. MACS2 was used for peak calling[20]. ChIPseeker (1.30.3) and ChIPpeakAnno (3.28.1) were used to annotate the peaks and find the overlapping peaks among different samples[43,44].

The MATCH tool of TRANSFAC (Version 2.0) was used for TFs prediction[45]. The sequences of ChIRP probes are listed in Supplementary Table 1. The ChIRP-seq data can be retrieved from the NCBI GEO database (GSE218931).

**Statistics and reproducibility**. Statistical analyses were performed using Graph-Pad Prism (version 9). All data are expressed as mean ± SEM and comprise ≥3 biologically independent replicates if not stated otherwise. All mice experiments were conducted at least twice with 5–11 mice in each group based on sample availability. Statistical significance was analyzed using a two-tailed unpaired Student's *t*-test. The threshold of statistical significance was set at $P < 0.05$.

**Reporting summary**. Further information on research design is available in the Nature Portfolio Reporting Summary linked to this article.

## Data availability
The uncropped western blot images for Figs. 2c, 3e, 5h, i are in Supplementary Figure 7. The RNAseq and ChIRP-seq data that support the findings of this study have been deposited in the NCBI GEO database (GSE218931, GSE218904, and GSE218905). All other data were available from the corresponding author on reasonable request.

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

## Acknowledgements

We would like to thank Drs. Arthur Riggs, Rama Natarajan, and other members at AR-DMRI for their discussion and suggestions. This study was supported partially by the George & Irina Schaeffer, the John Hench, and the Ella Fitzgerald Foundations and the National Institutes of Health grants (R01DK124627 to W.H., R01HL145170 and R01GM141096 to Z.B.C., and COH P30CA33572 to COH).

## Author contributions

W.H. conceptualized and designed the experiments and revised the manuscript. Z.F. and M.F. designed the experiments, analyzed data, wrote the manuscript, and performed the bulk of the experiments. L.J., Y.W., J.T., L.D., E.Z., and S.X. helped to collect data. D.Y., X.W., and Z.B.C. analyzed the ChIRP-seq data. Z.F., M.F., Z.B.C., and W.H. revised the manuscript.

## Competing interests

The authors declare no competing interests.
