## [Peer Review File · Communications Biology]

Reviewers' comments:

Reviewer #1 (Remarks to the Author):

In the current study, the author identified hepatic lncRNA Gm19619 as a sensitive response to ASG, and then they further probed its roles in the regulation of hepatic glucose and lipid metabolism. Finally, the author proposed that downregulation of Gm19619 may inhibit the expressions of G6pase and Pck1 to suppress hepatic gluconeogenesis, and inhibit leptin receptor to reduce lipid deposition. Overall, some of the findings are interesting. However, more experimental data are still needed to consolidated the key conclusion.

Comments:

- 1) The manuscript was focused on determining the roles of Gm19619 on hepatic gluconeogenesis. However, no direct evaluation of glucose production was performed in the study. So, the effects of Gm19619 overexpression and knockdown on glucose production in vivo should be evaluated by PTT in mouse models. Moreover, its effects on glucose production should also be determined in cultured hepatocytes.
- 2) What's the mechanism of Gm19619 on repression of gluconeogenic gene expressions? This should be further probed.
- 3) The authors proposed that Gm19619 may promote hepatic lipid deposition by inhibiting the expression of leptin receptor. So, it is interesting and important to verify whether knockdown of Gm19619 fails to correct the dysregulated glucose and lipid metabolism in db/db mice, which are leptin receptor deficient.
- 4) Overall, more mechanistical study are needed to consolidated the key findings and conclusion.

Reviewer #2 (Remarks to the Author):

The manuscript "Downregulation of hepatic lncRNA Gm19619 improves gluconeogenesis and lipogenesis following vertical sleeve gastrectomy in mice" by Fang, Fan and colleagues seek to understand how a VSG surgery regulated lncRNA Gm19619 regulates hepatic metabolism. Authors found that Gm19619 regulates gluconeogenic enzyme and leptin receptor expression. The topic is important and will be of interest to the readers of Communications Biology. The paper is clearly structured: the introduction lays out the need for this study, methods are carefully selected and suitable to test their hypothesis, conclusions are well-supported by the data, and discussions are thorough. As a result, the manuscript is a pleasant read.

Addressing a few minor concerns below will help to improve the manuscript:

1. In Fig 2. Effect of forced transcription of Gm19619 on hepatic gluconeogenesis: higher expression of gluconeogenic enzymes only indicates a potential for enhanced gluconeogenesis, but the gluconeogenesis process is also regulated by substrate availability. The golden method to prove elevated gluconeogenesis is by a hypothesis-driven tracing method with isotope-labeled metabolites. However, a pyruvate tolerance test in mice fasted overnight is an alternative with minimal technical requirements. The same applies to the knocking down study of Gm19619 in mice shown in Fig. 3.
2. A schematic showing when different assays were performed in Fig 2 and Fig 3 will be tremendously helpful.
3. In Fig 6: LepR western blotting showing the protein abundance tracks mRNA levels will strengthen the manuscript.
4. Formatting: Many citation numbers are under grey shade.

Can authors elaborate on the possible mechanism fasting and refeeding regulates Gm19619 in the discussion? And how is this related to VSG surgery's role in regulating hepatic lncRNA? Modulation of Gm19619 does not affect body weight, but VSG surgery is known to lead to significant weight

loss. Is it possible that other lncRNAs alone or together lead to body weight regulation? One future direction authors may consider is to knock down all the repressed lncRNAs shown in Fig. 1A to see whether they collectively deliver a body weight effect and a more substantial metabolic effect.

Reviewer #3 (Remarks to the Author):

In this manuscript, Fang Z et al., demonstrated a novel lncRNA Gm19619 function in obesity, which mediates the beneficial metabolic effects of bariatric surgery VSG. Hepatic Gm19619 upregulation promotes hepatic gluconeogenesis, whereas its ablation improves hepatic glucose/lipid metabolism, potentially through direct regulation on leptin receptor (Lepr) and other metabolic regulators. This is of course interesting to first report the metabolic function of lncRNA Gm19619 in hepatic metabolism, which provides a new and unexplored mechanism of metabolic interventions. However, several points need to be improved and revised to firmly establish the metabolic function of Gm19619.

Major points

1. To clearly evaluate gluconeogenic effect of Gm19619, additional metabolic phenotyping study such as Pyruvate tolerance test (PTT) or Clamp should be provided.
2. Does Gm19619 affect lipolysis in the liver and adipose tissues? In addition to the lipogenic gene expressions such as Srebp1 and Fasn, profiling of other genes in overall lipid metabolism (lipolysis, beta-oxidation etc) may be provided to interpret the metabolic phenotypes.
3. In this study, key finding includes that normalized Gm19619 potentially contributes to the metabolic benefits – decreased gluconeogenesis and restored leptin signaling. In the discussion part, additional paragraph describing how Gm19619 can be integrated into VSG biology (and control of hepatic leptin signaling) would be helpful. Does VSG also increase Lepr level or enhance hepatic leptin signaling?
4. Lines 205-207. Can Gm19619 directly regulate upstream regulators of G6pc and Pck1 transcriptions? which includes CREBP and FOXO1.

Minor points

1. In addition to Fig 1b, is Gm19619 level increased in HFD compared to normal chow, without surgery?
2. Line 113-114. This part needs some clarification. Authors mentioned there is no detectable morphological changes but, in fasted group of iWAT in Supple Fig 2d, adipocyte size seems smaller in AAV-Gm19619 group. Considering the size of adipocytes in eWAT and iWAT, HE image of AAV-GFP iWAT has been mis-placed.
3. Please clarify gene/protein expression analyses of CasRx-19x3 (including Fig 3f, 3g, 4c-e) were done in ad libitum (or short-term fasting before tissue collection). Also, Did authors analyze fasting serum NEFA levels in CasRx-19x3 mice (Fig 4d)?

Point-to-point response to reviewers' comments

Reviewer #1 (Remarks to the Author):

In the current study, the author identified hepatic lncRNA Gm19619 as a sensitive response to ASG, and then they further probed its roles in the regulation of hepatic glucose and lipid metabolism. Finally, the author proposed that downregulation of Gm19619 may inhibit the expressions of G6pase and Pck1 to suppress hepatic gluconeogenesis, and inhibit leptin receptor to reduce lipid deposition. Overall, some of the findings are interesting. However, more experimental data are still needed to consolidate the key conclusion.

Comments:

1) The manuscript was focused on determining the roles of Gm19619 on hepatic gluconeogenesis. However, no direct evaluation of glucose production was performed in the study. So, the effects of Gm19619 overexpression and knockdown on glucose production *in vivo* should be evaluated by PTT in mouse models. Moreover, its effects on glucose production should also be determined in cultured hepatocytes.

Thanks for the very insightful comments. As suggested, we performed the PTT assay *in vivo*, which confirmed the role of Gm19619 in glucose production in the liver. These data are now added in **Figure 2e** and **Figure 3d**. Additionally, we have also tried to use mouse primary hepatocytes for lncRNA overexpression/knockdown and glucose measurement, but were not able to achieve efficient overexpression/knockdown of Gm19619 in cultured hepatocytes due to technical difficulty.

Fig. 2. Adenovirus associated virus (AAV) mediated hepatic *Gm19619* forced transcription promoted hepatic gluconeogenesis in chow-fed mice. (a) Schema of the AAV vector for hepatic *Gm19619* transcription under the liver specific TBG promoter and the experimental timeline. GFP was used as the control. (b) The relative RNA level of *Gm19619*, *G6pc* and *Pck1* in AAV-Gm19619 (n=9) or AAV-GFP (n=7) infected mice liver after 16 h fasting and 4 h re-feeding by real time PCR. (c) Western blot analysis of protein levels of G6pc and Pck1 in AAV-Gm19619 or AAV-GFP mice liver

(n=3). (d) The blood glucose level of AAV-Gm19619 (n=18) and AAV-GFP (n=11) mice after overnight fasting. (e) GTT (n=11), ITT (n=11), and PTT (n=8) assay of the AAV-Gm19619 and AAV-GFP mice. * P < 0.05, ** P < 0.01, *** P < 0.001 and **** P < 0.0001.

Fig. 3. AAV-CasRx mediated *Gm19619* knockdown in high fat diet (HFD) fed mice significantly improved the mice glucose metabolism. (a) Schema of the AAV-EFS-CasRx vector for *Gm19619* knockdown and the experimental timeline. (b) The relative RNA levels of hepatic *Gm19619*, *G6pc* and *Pck1* in *Gm19619* knockdown (CasRx-19x3) (n=10) and control (CasRx-ctrl) (n=7) mice. (c) The fasting blood glucose levels in CasRx-19x3 mice (n=10) and CasRx-ctrl (n=7) mice. (d) The GTT, ITT and PTT assay of CasRx-19x3 (n=10, PTT n=8) and CasRx-ctrl (n=7) mice. (e) The western blot results of *G6pc* and *Pck1*

in CasRx-19x3 and control group mice livers (n=3). * P < 0.05, ** P < 0.01 and *** P < 0.001.

2) What's the mechanism of *Gm19619* on repression of gluconeogenic gene expressions? This should be further probed.

To further determine the underlying mechanism by which *Gm19619* regulates the gluconeogenic gene expression in the liver, we have performed deeper analysis on our ChIRP results. We found that *Gm19619* RNA could bind to the promoter region and genomic locus of *Foxo1* and increases *Foxo1* expression. We verified these new findings by real-time PCR and western blotting and the results are now added in **Figure 6**. Based on the data we have collected, we have included a proposed model for *Gm19619*-regulated gluconeogenic gene expression and glucose and lipid metabolism in the liver in **Figure 7**.

Fig. 6. *Gm19619* repressed the leptin receptor signaling and increased the *Foxo1* expression. (a) The relative mRNA levels of leptin receptor *Lepr* and *Foxo1* in *Gm19619* forced expressed mice (n=5-10). (b) Western blot analysis of protein levels of LEPR, FOXO1 and CREB1 in AAV-*Gm19619* or AAV-GFP mice liver (n=3). (c) The relative mRNA levels of *Lepr* and *Foxo1* in *Gm19619* knockdown mice (n=6-7). (d) The western blotting of LEPR, FOXO1 and CREB1 in the livers of CasRx-19x3 and control mice (n=3). * P < 0.05, ** P < 0.01 and *** P < 0.001.

Fig. 7. Schema showing the potential function of *Gm19619* in the liver. While VSG represses the transcription of hepatic *Gm19619*, both HFD and fasting induce *Gm19619* RNA. *Gm19619* may promote gluconeogenesis by activating the Foxo1-G6pc/Pck1 pathway, and facilitate the lipid accumulation by inhibiting leptin receptor signaling pathway.

3) The authors proposed that *Gm19619* may promote hepatic lipid deposition by inhibiting the expression of leptin receptor. So, it is interesting and important to verify whether knockdown of *Gm19619* fails to correct the dysregulated glucose and lipid metabolism in *db/db* mice, which are leptin receptor deficient.

We agree with the reviewer that it will be interesting and important to test whether *Gm19619* promotes hepatic lipid deposition by inhibiting *lepR*. However, besides liver, leptin receptor also plays key roles in hypothalamus and other tissues. Given that *Gm19619* is a liver-specific lncRNA, and *db/db* mice have *lepR* deficiency in the whole body, *Gm19619* knockdown in *db/db* mice is likely leading to failed metabolism protection. However, this cannot be attributed to liver-specific *Gm19619-lepR* pathway regulation of glucose and lipid metabolism. We will take reviewer's question and address it in our future study.

4) Overall, more mechanistical studies are needed to consolidate the key findings and conclusion.

We have performed further analysis and experiments as suggested by three reviewers to better characterize the potential mechanisms by which *Gm19619* regulates glucose and lipid metabolism in the liver. Specifically, we have identified that *Gm19619* RNA may also bind to the promoter region and genomic locus of *Foxo1* and promote *Foxo1* expression. In addition, the upstream regulator prediction of *Gm19619* in JASPAR database indicated that *Creb1* may bind to the promoter region of *Gm19619* and regulate its transcription. We have added the new results in **Figure 5g** and **Figure 6**. The more detailed mechanistic investigation on *Gm19619* will be our future research endeavor.

Fig. 5. ChIRP data analysis indicated Gm19619 binds to the chromatic DNA of leptin receptor and Foxo1. (a) The 2nd structure of Gm19619 RNA with minimal free energy as predicted on the RNAfold website. The locations of the 14 ChIRP probes were shown. (b) The genomic distribution of ChIRP pulldown DNA fragments with a pileup over 5 and longer than 200 bp. (c) Pathway enrichment analysis of genes with surrounding regions bound by *Gm19619* identified from ChIRP-seq. The gene list from overlapping peaks was used to run KEGG pathway analysis with IPA (Ingenuity Pathway Analysis). (d-e, g) ChIRP-seq signals indicate *Gm19619* binding along its own exon locus (d), the leptin receptor (e), and *Foxo1* gene locus (g). H3K27ac, H3K4me1, and H3K4me3 signals from ENCODE mouse liver ChIP-seq data were shown as control, mm10 as the reference genome. (f) The Circos plot showing the ChIRP enriched genes within the leptin receptor pathway.

Fig. 6. Gm19619 repressed the leptin receptor signaling and increased the Foxo1 expression. (a) The relative mRNA levels of leptin receptor *Lepr* and *Foxo1* in *Gm19619* forced expressed mice (n=5-10). (b) Western blot analysis of protein levels of LEPR, FOXO1 and CREB1 in AAV-Gm19619 or AAV-GFP mice liver (n=3). (c) The relative mRNA levels of *Lepr* and *Foxo1* in *Gm19619* knockdown mice (n=6-7). (d) The western blotting of LEPR, FOXO1 and CREB1 in the livers of CasRx-19x3 and control mice (n=3). * P < 0.05, ** P < 0.01 and *** P < 0.001.

Reviewer #2 (Remarks to the Author):

The manuscript “Downregulation of hepatic lncRNA Gm19619 improves gluconeogenesis and lipogenesis following vertical sleeve gastrectomy in mice” by Fang, Fan and colleagues seek to understand how a VSG surgery regulated lncRNA Gm19619 regulates hepatic metabolism. Authors found that Gm19619 regulates gluconeogenic enzyme and leptin receptor expression. The topic is important and will be of interest to the readers of Communications Biology. The paper is clearly structured: the introduction lays out the need for this study, methods are carefully selected and suitable to test their hypothesis, conclusions are well-supported by the data, and discussions are thorough. As a result, the manuscript is a pleasant read.

Addressing a few minor concerns below will help to improve the manuscript:

1. In Fig 2. Effect of forced transcription of Gm19619 on hepatic gluconeogenesis: higher expression of gluconeogenic enzymes only indicates a potential for enhanced gluconeogenesis, but the gluconeogenesis process is also regulated by substrate availability. The golden method to prove elevated gluconeogenesis is by a hypothesis-driven tracing method with isotope-labeled metabolites. However, a pyruvate tolerance test in mice fasted overnight is an alternative with minimal technical requirements. The same applies to the knocking down study of Gm19619 in mice shown in Fig. 3.

We are grateful for the positive feedback and valuable suggestions from the reviewer. We have taken the reviewer’s suggestion and performed the pyruvate tolerance test (PTT), which confirms the gluconeogenic effect of Gm19619 in the liver. The results have now been added in **Figure 2e** and **Figure 3d**. (please also see reviewer#1’s 1st Q&A)

2. A schematic showing when different assays were performed in Fig 2 and Fig 3 will be tremendously helpful.

This is an excellent suggestion. We have made schematic drawings showing the timelines of different assays and included them in **Figure 2a** for gain-of-function and **Figure 3a** for knockdown of Gm19619.

3. In Fig 6: LepR western blotting showing the protein abundance tracks mRNA levels will strengthen the manuscript.

Accordingly, we have performed western blotting for LepR and included these data in **Figure 6c** and **6d**.

Fig. 6. Gm19619 repressed the leptin receptor signaling and increased the Foxo1 expression. (a) The relative mRNA levels of leptin receptor *Lepr* and *Foxo1* in *Gm19619* forced expressed mice (n=5-10). (b) Western blot analysis of protein levels of LEPR, FOXO1 and CREB1 in AAV-Gm19619 or AAV-GFP mice liver (n=3). (c) The relative mRNA levels of *Lepr* and *Foxo1* in *Gm19619* knockdown mice (n=6-7). (d) The western blotting of LEPR, FOXO1 and CREB1 in the livers of CasRx-19x3 and control mice (n=3). * P < 0.05, ** P < 0.01 and *** P < 0.001.

4. Formatting: Many citation numbers are under grey shade.

Thanks for the critical review and we apologize for our oversight. We have checked all citations carefully and made sure they are formatted appropriately in the revised manuscript.

5. Can authors elaborate on the possible mechanism fasting and refeeding regulates Gm19619 in the **discussion**? And how is this related to VSG surgery's role in regulating hepatic lncRNA? Modulation of Gm19619 does not affect body weight, but VSG surgery is known to lead to significant weight loss. Is it possible that other lncRNAs alone or together lead to body weight regulation? One future direction authors may consider is to knock down all the repressed lncRNAs shown in Fig. 1A to see whether they collectively deliver a body weight effect and a more substantial metabolic effect.

We appreciate the reviewer's suggestions on improving the Discussion. We have now included in the discussion all these points, including 1) more information regarding the Gm19619 regulation; and 2) potential roles of other lncRNAs in addition to Gm19619 underlying the salutary effect of VSG surgery in regulating body weight and other metabolic effects. We will test the effects to knockdown all the VSG-repressed lncRNAs on body weight and metabolic function in our future studies.

Reviewer #3 (Remarks to the Author):

In this manuscript, Fang Z et al., demonstrated a novel lncRNA Gm19619 function in obesity, which mediates the beneficial metabolic effects of bariatric surgery VSG. Hepatic Gm19619 upregulation promotes hepatic gluconeogenesis, whereas its ablation improves hepatic glucose/lipid metabolism, potentially through direct regulation on leptin receptor (*Lepr*) and other metabolic regulators. This is of course interesting to first report the metabolic function of lncRNA Gm19619 in hepatic metabolism, which provides a new and unexplored mechanism of metabolic interventions. However, several points need to be improved and revised to firmly establish the metabolic function of Gm19619.

Major points

1. To clearly evaluate gluconeogenic effect of Gm19619, additional metabolic phenotyping study such as Pyruvate tolerance test (PTT) or Clamp should be provided.

We are extremely grateful for the reviewer's recognition of the novelty of our study and all the helpful comments. Per this excellent suggestion, we have performed PTT, which confirmed the gluconeogenic effect of Gm19619 in the liver. The data have now been added in **Figure 2e** and **Figure 3d**. (please also see reviewer#1's 1st Q&A)

2. Does Gm19619 affect lipolysis in the liver and adipose tissues? In addition to the lipogenic gene expressions such as *Srebp1* and *Fasn*, profiling of other genes in overall lipid metabolism (lipolysis, beta-oxidation etc) may be provided to interpret the metabolic phenotypes.

To address this insightful question, we have measured the expression of genes in lipid metabolism in adipose tissues. The results showed no significant difference of these gene expressions in adipose tissues, suggesting Gm19619 does not affect lipolysis in adipose tissues. We have also measured the expression of additional genes in lipid metabolism in the liver, including *Acs1*, *Atsl*, *Hsl*, *Cpt1a* and *Cpt1b*. These new data have been included in the new **Figure 4f**.

Fig. 4. AAV-CasRx mediated *Gm19619* knockdown in HFD-fed mice significantly improved the mice lipid metabolism. (a) The body weight of *Gm19619* knockdown (CasRx-19x3) (n=11) and control (n=8) mice. (b) Representative images of liver sections by H&E and oil red o staining, respectively. (c) The serum level of triacylglycerol (TG) in CasRx-19x3 (n=11) and CasRx-ctrl (n=6) mice. (d) The serum level of NEFA in CasRx-19x3 (n=11) and CasRx-ctrl (n=8) mice. (e) The relative mRNA levels of some

lipogenesis genes and Cd36 in CasRx-19x3 (n=11) and CasRx-ctrl (n=8) mice. (f) The relative mRNA levels of several lipolysis and beta oxidation genes in CasRx-19x3 and CasRx-ctrl mice (n=8). * P < 0.05 and *** P < 0.001.

3. In this study, key findings include that normalized Gm19619 potentially contributes to the metabolic benefits – decreased gluconeogenesis and restored leptin signaling. In the discussion part, **additional paragraph describing how Gm19619 can be integrated into VSG biology (and control of hepatic leptin signaling)** would be helpful. Does VSG also increase *Lepr* level or enhance hepatic leptin signaling?

Thanks for the helpful comments. We have added more updates in the discussion section. 1) The circulating leptin level was found to be reduced after VSG, and yet VSG can still reduce fat mass and improve glucose tolerance in ob/ob mice in the absence of the leptin signaling. In this regard, in addition to the repression of Gm19619 in the liver, other signaling pathways may also have significant impact on lipid metabolism after the VSG surgery. 2) We found that Gm19619 also bound to the promoter of the *Foxo1* and activated its expression to modulate the hepatic glucose metabolism. Thus, the downregulation of Gm19619 by VSG may contribute to the improved hepatic glucose metabolism in part through the *Foxo1* pathway.

4. Lines 205-207. Can Gm19619 directly regulate upstream regulators of *G6pc* and *Pck1* transcriptions? which includes CREBP and FOXO1.

This is an excellent question. Partially as anticipated, we found that Gm19619 RNA could promote FOXO1 expression (Fig. 6). Moreover, ChIRP data show that Gm19619 can bind to the promoter region and genomic locus of *Foxo1*. In contrast, Gm19619 does not seem to regulate the expression of CREBP. However, based on the upstream regulator prediction by JASPAR database, CREBP may bind to the promoter of Gm19619 and regulate the transcription of Gm19619 (Supplemental Fig. 6).

Fig. 6. Gm19619 repressed the leptin receptor signaling and increased the Foxo1 expression. (a) The relative mRNA levels of leptin receptor *Lepr* and *Foxo1* in *Gm19619* forced expressed mice (n=5-10). (b) Western blot analysis of protein levels of LEPR, FOXO1 and CREB1 in AAV-Gm19619 or AAV-GFP mice liver (n=3). (c) The relative mRNA levels of *Lepr* and *Foxo1* in *Gm19619* knockdown mice (n=6-7). (d) The western blotting of LEPR, FOXO1 and CREB1 in the livers of CasRx-19x3 and control mice (n=3). * P < 0.05, ** P < 0.01 and *** P < 0.001.

Supplemental Fig. 6. The prediction of potential transcription factors binding around Gm19619 promoter region by JASPAR. Note that the putative promoter region of Gm19619 may be bound by CREB1, PPAR α , FXR, or other TFs known to regulate glucose metabolism in fasted and fed states.

Minor points

1. In addition to Fig 1b, is Gm19619 level increased in HFD compared to normal chow, without surgery?

Yes, HFD feeding increased the transcription of Gm19619 without surgery.

2. Line 113-114. This part needs some clarification. Authors mentioned there is no detectable morphological changes but, in fasted group of iWAT in Supple Fig 2d, adipocyte size seems smaller in AAV-Gm19619 group. Considering the size of adipocytes in eWAT and iWAT, HE image of AAV-GFP iWAT has been mis-placed.

We apologize for the lack of clarity and have examined the H&E images carefully. The H&E image of AAV-GFP iWAT has been updated in the new **Supplemental Figure 2d** to best represent our observation with AAV-Gm19619.

Supplemental Fig. 2. Forced transcription of lncRNA *Gm19619* did not affect the body weight and lipid metabolism in chow-fed mice. a) The mouse body weight when the GTT and ITT test were performed after AAV was injected 4 week (n=14) and 5 weeks (n=20). b) Body weight of mice from fasted (n=7) and re-fed (n=7) AAV-GFP ; fasted (n=9) and re-fed (n=11) AAV-Gm19619. c) liver /body weight ratio (n=5-10). d) Representative images of the liver, iWAT and eWAT sections stained with indicated reagents; scale bar, 100 μ m. e). Relative mRNA levels of genes involved in lipogenesis, (n=5-10). *P < 0.05; ns: no significance.

3. Please clarify gene/protein expression analyses of CasRx-19x3 (including Fig 3f, 3g, 4c-e) were done in ad libitum (or short-term fasting before tissue collection). Also, did authors analyze fasting serum NEFA levels in CasRx-19x3 mice (Fig 4d)?

Indeed, the gene/protein expression analyses of CasRx-19x3 were done ad libitum. We did not analyze fasting serum NEFA levels in CasRx-19x3 mice.

REVIEWERS' COMMENTS:

Reviewer #1 (Remarks to the Author):

The authors sufficiently addressed all of my previous concerns by performing new experiments, which made the conclusion more solid. There is no further comment.

Reviewer #2 (Remarks to the Author):

Thank you for addressing all the concerns raised in the initial review. The revised manuscript is technically sound and will offer great value to the field of the function of lncRNA Gm19619 in liver metabolism.

Reviewer #3 (Remarks to the Author):

This is the very unique story on the novel function of lncRNA, which links bariatric surgery with the phenotypic changes in the liver and adipose tissues. The authors are very responsive to the reviewer's comments. All concerns are resolved and suggestions are included in the manuscript.